# Effective Dosage of Oral Vancomycin in Treatment for Initial Episode of *Clostridioides difficile* Infection: A Systematic Review and Meta-Analysis

**DOI:** 10.3390/antibiotics8040173

**Published:** 2019-10-01

**Authors:** Chia-Yu Chiu, Amara Sarwal, Addi Feinstein, Karen Hennessey

**Affiliations:** 1Department of Internal Medicine, Lincoln Medical Center, New York, NY 10451, USA; sarwala@nychhc.org; 2Department of Internal Medicine, Section of Infectious Diseases, Lincoln Medical Center, New York, NY 10451, USA; feinstea@nychhc.org (A.F.); karen.hennessey@nychhc.org (K.H.)

**Keywords:** *Clostridioides difficile*, *Clostridium difficile*, vancomycin, fecal vancomycin concentration, escalate dosage

## Abstract

Background: Oral vancomycin is a first line treatment for an initial episode of *Clostridioides difficile* infection. However, the comparative efficacy of different dosing regimens is lacking evidence in the current literature. Methods: We searched PubMed, Embase, Cochrane Library, and ClinicalTrials.gov. from inception to May 2019. Only articles published in English are reviewed. This meta-analysis compares the effects of low dose oral vancomycin (<2 g per day) versus high dose vancomycin (2 g per day) for treatment of initial *Clostridioides difficile* infection. Results: One randomized controlled trial and two retrospective cohort studies are included. A total of 137 patients are identified, 53 of which were treated with low dose oral vancomycin (39%) and 84 with high dose oral vancomycin (61%). There is no significant reduction in recurrence rates with high dose vancomycin compared to low dose vancomycin for treating initial episodes of non-fulminant *Clostridioides difficile* infection ((odds ratio (OR) 2.058, 95%, confidence interval (CI): 0.653 to 6.489). *Conclusions*: Based on limited data in the literature, low dose vancomycin is no different than high dose vancomycin for treatment of an initial episode of *Clostridioides difficile* infection in terms of recurrence rate. Additional large clinical trials comparing the different dosages of vancomycin in initial *Clostridioides difficile* infection are warranted.

## 1. Introduction

*Clostridioides difficile* infection (CDI) is caused by a disruption of intestinal microbiota due to antimicrobial therapy and an exposure of to *Clostridioides difficile* spores, such as fecal oral transmission from environmental surfaces, shared instrumentation, infected roommates, and iatrogenically [1]. In 2017, guidelines for CDI treatment issued by the Infectious Diseases Society of America (IDSA) and the Society for Healthcare Epidemiology of America (SHEA) recommended vancomycin 125 mg orally given 4 times per day for 10 days or fidaxomicin 200 mg given twice daily for 10 days for treatment of an initial episode. Although severe CDI is defined by leukocytosis of ≥15,000 cells/mL or a serum creatinine level >1.5 mg/dL, severity of initial episode of CDI does not change the aforementioned dosing regimen [2]. Current guidelines recommend one to escalate vancomycin to 500 mg 4 times per day and consider adding intravenous metronidazole in an initial episode of fulminant CDI. Fulminant CDI was defined as hypotensive, shock, ileus, or megacolon [2].

In 1893, pseudomembranous colitis (PMC) was first described by Fenney. *Clostridioides difficile* was first described in 1935 as a normal intestinal flora of newborn infants. In the 1950s, PMC became known as an antibiotic complication, and was thought to be primarily encountered by surgeons as “postoperative diarrhea” or “antibiotic-associated colitis”. In the 1970s, this was thought to be “clindamycin-induced enterocolitis” [3]. Vancomycin was used as a treatment because *Staphylococcus aureus* was the suspected pathogen during that era and based on the findings of experiments using hamster models [4,5]. In 1974, Green studied penicillin-induced death for guinea pigs and hamsters [6]. He concluded that it was an “activation of a latent virus by penicillin” which caused this infection, and later it was proven that it was the *C. diff* cytotoxin. In 1986, vancomycin was approved by the Food and Drug administration (FDA) for treatment of CDI and was the first and only medication for CDI [3,7]. Two concerns of using vancomycin at this time were (1) cost and (2) colonization with vancomycin-resistant enterococcus. Metronidazole was studied and proved to be an effective treatment for *C. diff*, similar to vancomycin [3,8]. Therefore, in the 2010 IDSA guidelines, metronidazole alone was listed as a first line therapy for mild to moderate initial episodes of CDI and vancomycin was reserved as a treatment for severe initial episodes [9].

In the 1970s, the most common regimen for an initial episode of CDI was vancomycin dosed at 500 mg given orally 4 times per day [10,11,12]. Some retrospective studies that were performed at a single center did show that vancomycin <2 g orally per day may be as effective as a total vancomycin dose of 2 g per day (500 mg 4 times per day) [10,11,12]. However, there was a lack of large, double blinded, multi-center studies to provide strong evidence to support that finding. During that decade and into the next, clinicians were attempting to decrease the cost of treatment by decreasing the overall daily dosing of vancomycin, and were concerned about systemic absorption of oral vancomycin in patients with renal insufficiency [12,13]. In the 1980s, the cost 125 mg of vancomycin given orally 4 times per day was $16 while treatment with 500 mg given orally 4 times per day was $63 [12]. Nowadays, the cost of vancomycin is negligible compared with cost to treat recurrent CDI or length of stay in hospital.

In the most recent set of guidelines in 2017, fidaxomicin had replaced metronidazole as a recommended therapy [2,14,15,16,17]. Vancomycin still remains as one of the first line therapies since the 2010 IDSA guidelines, but it is now acceptable for use in non-severe CDI [9,18]. The 2017 IDSA guidelines highlight vancomycin 125 mg given 4 times daily for 10 days (strong recommendation) in an initial episode of CDI, however, the optimal dosage of vancomycin has not been well elucidated for an initial episode of CDI as the comparative trials were not based on high-quality evidence.

Theoretically, the fecal pharmacokinetics of orally administered vancomycin may give us a better idea about the appropriate dosage of vancomycin. However, the studies assessing vancomycin concentration in stools are scant. A study in 2010, which was comprised of 15 patients (including nine patients with confirmed CDI) was designed to address three different vancomycin dosing regimens. Nine patients were administered 125 mg of vancomycin every 6 h, four patients were administered 250 mg every 6 h and two patients were administered 500 mg every 6 h [19]. All patients reached 100 times higher than minimum inhibitory concentration (MIC) 90, with only one patient who was administered 125 mg of vancomycin every 6 h having a relatively lower fecal vancomycin level, although it was still above MIC90. That study also showed higher stool frequency (more than 4 times per days) had a lower fecal vancomycin level. They hypothesized that by having higher frequency, the stool may have a “dilution effect”. A second study done in 2015, which was comprised of 15 patients who received vancomycin given 125 mg 4 times per day. Fecal vancomycin concentrations during the course (day 3–5) of therapy did not differ with either stool consistency or frequency [20]. At the end of treatment (day 10–13), high vancomycin concentration with high stool frequency was found. They explained that the higher frequency of the stool had less time to be diluted. That study also showed fecal concentration did not associate with cure or treatment failure. The limitations of these two studies are that they have a small sample size, and inconsistent conclusions [19,20].

A recurrence of CDI is defined as an episode of symptom onset and positive assay results following an episode with positive assay results in the previous 2–8 weeks [2]. The different dosage of vancomycin still has not been evaluated as a factor associated with recurrence. The present article aims to explore the effectiveness of the dosage of vancomycin in regards to recurrent rate.

## 2. Methods

### 2.1. Search Strategy and Selection Criteria

This study was conducted in accordance with the PRISMA guidelines on reporting systematic reviews [21]. Clinical studies reporting outcomes of patients with *Clostridioides difficile* were screened. All study types except case reports, case series, and conference abstracts were considered. PubMed, Embase, Cochrane Library, and ClinicalTrials.gov were searched from the from inception record to May 2019 using the following search protocol: ((*Clostridium difficile*) AND (vancomycin) AND (dose OR dosage)). The searching strategy was identically applied to all databases. Only articles published in English were reviewed. We only included articles that focused on initial episode of CDI. Studies directly comparing clinical resolution and recurrence by different dosing of vancomycin were included. If the original publication did not contain sufficient information about patient outcome, we requested additional data from the first author or corresponding authors by e-mail. Studies were excluded if they did not report outcomes associated with each antibiotic agent, or if the authors were unable to provide such data upon request. In this study, we define low dose vancomycin treatment as a patient who receives less than 2 g of oral vancomycin per day and we defined high dose vancomycin treatment as a patient who receives 2 g of oral vancomycin per day.

### 2.2. Data Extraction and Bias Assessment

Two reviewers (C.C. and A.S.) independently evaluated all eligible articles. We recorded the first author, year, sample size, number, and type of treatment arms, and participant characteristics. Data for resolution of symptoms and recurrence rate were extracted from the published article, or provided by authors on request.

### 2.3. Statistical Analysis

Clinical resolution and odds ratio (OR) of recurrence in the low dose vancomycin group compared with the high dose vancomycin group comprised the outcome. A random effects model was employed to pool individual OR; all analyses were performed using comprehensive meta-analysis (CMA) software, version 3 (Biostat, Englewood, NJ, USA).

Between-trial heterogeneity was determined by using I2 tests; An I2 > 50% was considered as statistically significant heterogeneity funnel plots and Egger’s test were used to examine potential publication bias. Statistical significance was defined as *p*-values < 0.05, except for the determination of publication bias, that employed *p* < 0.10.

## 3. Results

We retrieved 1069 non-duplicated citations for a review of their title and abstracts, and included 10 articles for meticulous evaluation after eliminating references based on our inclusion criteria (Figure 1). We contacted the corresponding authors of three studies to request additional data [22,23,24] and received replies from the authors of two studies [22,24]. In addition, one study is a conference poster [23]. Of those two studies, one author replied that the requested recurrence rate data was not available but provided additional detailed standard deviation data according to each group of antibiotic agents [22]. Characteristics of the studies included are summarized in Table 1. In total, the study included one randomized controlled trial [12] and three retrospective cohort studies [11,22,24]. One retrospective cohort study was designed to compare escalated vancomycin dose to 500 mg 4 times per day in patients who failed to respond to conventional dose of vancomycin. In that retrospective cohort study, there was a group of patients who received high dose vancomycin 500 mg given 4 times per day (14 patients) for an initial episode CDI [22].

The pooled OR of recurrence in the low dose vancomycin arm compared with the high dose vancomycin arm was 2.058 (95% CI: 0.653 to 6.489, *p* = 0.958), indicating a reduced incidence of recurrence following high dose vancomycin (Figure 2, Table A2). Regarding the heterogeneity of OR, the I2 was less than 0.01%. The Egger’s test revealed no significant publication bias regarding the OR of recurrence (*p* = 0.500).

## 4. Discussion

This study includes reports from 1981, 1989, 2013, and 2018. Given that only four studies are available in a time frame of nearly four decades, this shows that there is a lack of evidence about efficacy of the dosing of vancomycin in treatment of initial CDI. We combined three studies and found a trend towards a reduced risk of recurrence in the high dose arm compared with the low dose arm but the outcome was not statistically significant. Current research shows that fidaxomicin (200 mg given twice daily) is noninferior to low dose vancomycin (125 mg given 4 times daily) and fidaxomicin was associated with a significantly lower rate of recurrence of CDI [17]. Nevertheless, high dose vancomycin (500 mg 4 times given per day) was not used to compare with fidaxomicin directly. In that study clinical cure was defined by the resolution of diarrhea (i.e., three or fewer unformed stools for 2 consecutive days). In our opinion, high dose vancomycin needs to be compared with either fidaxomicin or low dose vancomycin. That will provide a clearer idea about the most efficacious dose of vancomycin for treatment of initial CDI.

In one retrospective study published by Cunha et al., they escalated vancomycin from 250 mg to 500 mg 4 times per day when the patient did not have a clinical response in three days [22]. They saw improvement when escalating the dose. They mentioned that according to their three decades of experience, 125 mg 4 times per days often failed to achieve rapid clinical improvement of diarrhea. When to escalate, how to escalate, and the maximal oral dosage of vancomycin that should be administered remains unclear and more studies are needed to help guide clinicians when they treat CDI. In the Cunha study, some of the treatments included concomitant administration of metronidazole with vancomycin. Another question raised is regarding metronidazole as a confounding factor when administered with different doses of vancomycin in CDI, however one previous study showed combination of metronidazole and vancomycin increased the risk of candidemia [25].

Red man syndrome, renal toxicity, and ototoxicity in oral vancomycin are controversial [26,27,28,29]. Detectable serum concentrations of vancomycin were noted after oral administration in dialysis patients, and impaired gastrointestinal mucosa in a stem cell transplant patient [27,30]. Risk factors included vancomycin dose more than 500 mg daily, treatment more than 10 days, vancomycin enema, preexisting gastrointestinal inflammation, and intensive care unit admission [31]. However, in one study consisting of eight pediatric patients, there was no report of a detectable serum vancomycin level or adverse effect [32].

We are unable to combine all the clinical cure data among four studies because these studies did not use the same criteria for a clinical cure (Table A1). Two studies defined this as less than 4 formed stools per day [11,12]. One study defined a clinical cure as resolution of diarrhea for ≥48 h without the development of a complication, including colectomy, colonic perforation, ileus, and toxic megacolon [24]. One study defined it as soft/formed stool without watery stool [22]. Although all aforementioned studies measured clinical cure as a primary endpoint, inconsistency in the definition makes it difficult to be compared across studies. These papers included exclusively symptomatic diarrhea patients as none of the patients included were displaying asymptomatic *C. diff* colonization. We also found that there was no unifying definition of a clinical cure to allow clinicians to measure CDI objectively. Further guidelines are needed to establish a uniform definition of a clinical cure.

Vancomycin capsules were approved for treatment of CDI in 1986 [7]. At that time, the estimated cost for 10 days of vancomycin at 125 mg dosage was $160 while a 500 mg dosage was $640 [12]. In 2017, the estimated cost of 10 days of vancomycin at 125 mg dosage was $2640 while a 500 mg dosage was $9760 [7]. High dose vancomycin is not only costly but also has the potential for detectable serum concentrations in certain patient population. Therefore, it would be worthwhile for future research to be done in this area.

There are several limitations of the present study. First, we were unable to obtain additional data regarding clinical outcomes from two studies [10,23]. Second, only three articles were able to be thoroughly analyzed as one study did not have recurrence data available from author [22]. Third, we were unable to perform subgroup analysis based on the severity of CDI. Further studies are needed to explore efficacy of different dosages of vancomycin stratified by disease severity as well as the validation of severity criteria.

## 5. Conclusions

In conclusion, to the best of our knowledge this is the first study regarding the clinical cure and recurrence rate of first episode of CDI in patients receiving different dosages of oral vancomycin. Although current guidelines recommend 125 mg of oral vancomycin given 4 times per day as one of the mainstay treatments in initial episodes of CDI, future studies need to be done to focus on different dosages of vancomycin that may decrease length of stay or recurrence rate. Compared with the cost of length of stay, the cost of high dose oral vancomycin is negligible. On the other hand, all the papers included in our study used different definitions of a clinical cure, which make us unable to utilize these data to analyze or compare the clinical effect.

## Reference

## Figures and Tables

**Figure 1 antibiotics-08-00173-f001:**
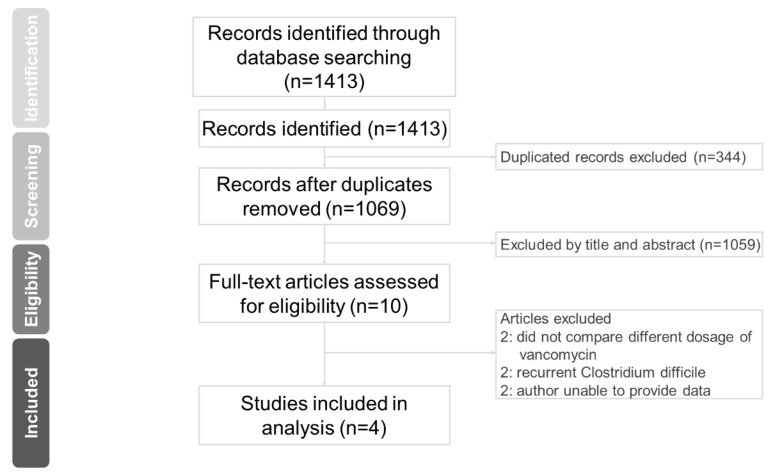
PRISMA Flow Diagram for study selection.

**Figure 2 antibiotics-08-00173-f002:**
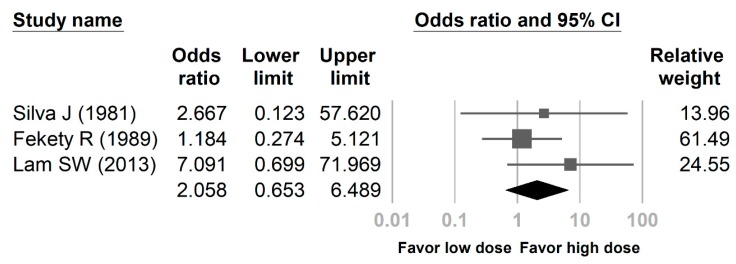
Forest plot presenting the odds ratio (OR) of recurrence in patients with initial episodes of *Clostridioides difficile* infection treated with low dose vancomycin versus high dose vancomycin.

**Table 1 antibiotics-08-00173-t001:** Characteristics of the four articles included in this study.

Author, Year, Country, Reference	Study Design, Period	Sample Size	Mean Age (Years)	Outcomes Examined	Recurrence Definition
Silva et al., 1981, USA, [11]	Retrospective cohort, NA	Low dose (less than 1g per day): 4 High dose (500 mg given 4 times per day): 9	NA	Decrease fever, abdominal pain, less than 4 formed stool/day	Within 42 days
Fekety et al., 1989, USA. [12]	RCT, NA	Low dose (125 mg given 4 times per day): 24 High dose (500 mg given 4 times per day): 22	Low dose: 56 High dose: 52	Cessation of diarrhea, duration of therapy, post treatment carriage rate, follow up (2 to 6 weeks)	Within 42 days
Lam et al., 2013, USA, [24]	Retrospective cohort, 2006–2011	Low dose (125 mg given 4 times per day): 16 High dose (500 mg given 4 times per day): 32	Low dose: 65 High dose: 69	Clinical cure, recurrence, length of stay, complication, mortality	Within 30 days
Cunha et al., 2018, USA, [22]	Retrospective cohort, 2015–2016	Conventional dose: 113 ‡ High dose (500 mg given 4 times per day): 14	Conventional dose: 69.6 High dose: 64	Clinical resolution, treatment duration	NA #

RCT: randomized controlled trial. NA: not available. ‡ 125 mg given 4 times per day (5 patients) and 250 mg given 4 times per days (108 patients) # Recurrence rate data was not available. Did not include in meta-analysis.

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
