# Peer review of "Effective Dosage of Oral Vancomycin in Treatment for Initial Episode of Clostridioides difficile Infection: A Systematic Review and Meta-Analysis"

_antibiotics, 2019, doi:10.3390/antibiotics8040173_

Round 1

Reviewer 1 Report

In this manuscript entitled “Effective dosage of oral vancomycin in treatment for initial episode of Clostridoides difficile infection: A systemic review and meta-analysis” the authors aim to explore the effectiveness of the dosage of vancomycin in regards to initial cure and recurrent rate of CDI. In general, this paper is listed as a “review” but the limited literature is included in this review, therefore this is misleading to the reader. Additionally, the meta-analysis only was able to evaluate 4 papers. It is hard to extrapolate information from 4 papers that different not only in dose of vancomycin administered but also definitions of clinical outcomes for patients. In its current state this paper is better suited as a short communication and does not meet the reader’s expectations for a “review” on this topic.

Major revisions:

1.  Need to include a thorough review of medical management of initial CDI.

2. Need to reexamine if this data is suited for a meta-analysis given the heterogeneity between the 4 studies included. I am concerned that since the primary endpoint varied  by study that this cannot be used in the meta-analysis section as a point of comparison.

Minor Revisions:

Line 39: Be specific that this is hospital stays

Lines 52-54: Discusses costs but these are VERY outdated

Lines 57-60: This sentence is poorly worded and needs to be revised

Line 76: S. boulardii is not a drug and this needs to be corrected

Lines 96-100: These sentences are poorly worded and need to be revised

Figure 1: Under excluded articles it should read “data” not date

Table 1: Needs a column for the dose of vancomycin given since this varied across studies

Figure 2 and 3: Have wrong name of pathogen and this needs to be corrected.

Lines 166-172: Please list out the total cost of the medications.

Reviewer 2 Report

I applaud you for trying to answer the important question of the optimal dosing regimen of vancomycin for CDI. I think the manuscript could be excellent with collaboration with a researcher with meta-analysis experience and a careful proofread. A few suggestions to potentially improve the quality of this manuscript:

Abstract

Line 13- This sentence doesn’t seem to capture the major impetus for conducting this meta-analysis, which is the lack of data supporting dosing regimens for vancomycin as a treatment for initial episodes of CDI.

Line 16- There appears to be a discrepancy with the dates here and in the body of the paper.

Line 16- The sentence that begins with “Compare the effects” is not a full sentence.

Lines 23-26- The wording here is hard to follow, and I think it should be more clear that there was not a statistically significant difference given the low sample size with this outcome

Line 28- Suggest rewording the last sentence

Introduction

Lines 34-39- Are the causes and financial burden of CDI the most pertinent pieces of information to be in the first sentences of the background of this manuscript? If anything, I would think that the point to make would be that despite the prevalence of this disease state, we still don’t have strong data supporting the dosing regimen for one of our first-line agents.

Line 41-44- I would suggest providing more details about the level of evidence and rationale in the guidelines for selecting this dose. I would also like to see more information about pharmacokinetics and the achievable concentrations of vancomycin in the colon to inform rational dose selection.

Line 45-46- I wasn’t practicing in the 1970s to know if this was true, but I’m not sure that the references here support the assertion about the most common regimen at that time. I’m also questioning the value of this timeline to the paper- are you trying to illustrate how changes in the dosing regimen over time are being made without strong evidence?

Line 55- Is the information about other first line therapies relevant to this paper?

Line 66- Is the information about reducing recurrence rates significant here? It doesn't appear that this was information you collected as potentially confounding in the analysis

Method- [probably should be methods?]

Lines 73-82- It appears that other inclusion criteria are missing here, including that it had to be the initial episode of CDI and the study had to directly compare clinical outcomes of the two different dosing regimens

Lines 96, 98- Is this “we abbreviated this in the abstract” an internal note to the authors?

Line 96- I think you need to describe more about how you calculated the standardized mean difference. It doesn’t really make sense to me that you pooled individual SMD from the studies.

Line 126- The definitions used in the study should probably be in the methods, not the results

I think you need to define symptom improvement and recurrence a little bit more- was it typically defined as the percentage of patients with cure at a specific time point, or was it the number of days to clinical cure?

Some experts (notable the Cochrane Handbook) recommend against using bias scores like Jadad. It seems important to highlight the risk of bias given that most were retrospective.

Results

Line 135- Why would the SMD about symptom improvement allow us to draw conclusions about the number of days to clinical cure? As noted above, I may be confused about what this endpoint was. Is it days to clinical cure, or the number of patients who had clinical cure at a pre-specified time point? I think you may want to report the actual raw values for cure and recurrence somewhere in the study instead of the standard difference and OR only so that the reader has some context.

Discussion

Line 152- Not sure that the discussion about fidaxomicin is adding anything here, unless you are suggesting that it might not be noninferior to higher dose vanco? That may be a stretch based on the information you’ve provided.

Lines 158-162- Should the definitions from the various studies, and your justification for combining these heterogeneous endpoints, be included in the methods instead of the discussion? Do you feel like these endpoints are so different that they shouldn’t have been combined in a meta-analysis?

Line 171- You haven’t provided any evidence or suggestion to this point that a higher dose of vancomycin could cause more severe side effects. I’m not sure that it’s true with an antibiotic that is unlikely to be systemically absorbed in meaningful quantities (patients with renal dysfunction excluded)

What about the way that CDI was diagnosed in the studies? Was this consistent among the studies and was there a possibility that included patients may have been asymptomatic and colonized vs infected?

I’m concerned that you still have a very small number of included patients, particularly in the high dose arms, and a wide confidence interval for most of your endpoints, but still draw conclusions from this data without mentioning power and statistical significance as a major limitation

Do you feel like concurrent treatment with metronidazole (particularly in the Cunha study) needs to be addressed in terms of impact on outcomes? Any other potentially confounding variables that should be addressed as it relates to recurrence or cure?

Conclusion- The last sentences are not addressing the conclusion of the paper

Figure 1- Typos- did not compared, Unable to provide date

Figure 1- It’s not clear to me the distinction between studies included in qualitative synthesis and studies included in meta-analysis? If there was a qualitative analysis, you could have included some of the other studies that were excluded because of lack of data?

Table 1- Would include information about severity of CDI and severity of illness; I personally don’t find the funding column particularly enlightening for this study. I think it would help the readers if you simply put the doses in the table instead of having multiple footnotes. I’m not sure how many readers will be able to interpret the quality assessment as a number without context (is a high number good or bad?).

Round 2

Reviewer 2 Report

Overall- I appreciate the changes that were made to the manuscript, including the removal of the clinical cure meta-analysis due to the variable definitions included in the various studies and the addition of fecal PK data. I can see that a significant amount of work has been done to improve the quality of the manuscript. I do wonder if you still want to include the studies you found that compared clinical cure rates in a more qualitative way, since it's still an important part of this discussion even if it can't be combined in meta-analysis? Perhaps a table comparing the different studies and the definitions of cure that they used. I would also point out that I believe some of the original review comments may have been misinterpreted, which may explain what may appear to be discordant recommendations below.

Title- I still think the title should reflect that this was a review, since it currently doesn't say what the methodology was.

Abstract:

Line 12- ‘mainstay’ seems vague, should you just say first-line?

Line 12- needs to say either “an initial episode” or “initial episodes”

Line 13-14- The two parts of this sentence seem to be saying the same thing, and I think perhaps this should be more specific in that it’s the comparative efficacy of different dosing regimens that’s missing from the literature

Methods- Is this still considered a systematic review and meta-analysis? I think I'm confused about why it's being removed if you still performed the meta-analysis of one outcome.

Line 19- 4 studies were identified, but you didn’t include them all in the meta-analysis

Lines 24-25- would probably reword this  (maybe there is no significant reduction in recurrence rates with high dose vanco compared to low dose vanco for treating initial episodes of non-fulminant CDI?)

Intro

Line 32-33- It seems to me that these two first sentences are linked. Patients need to be BOTH exposed to CD spores in the environment AND also have some kind of disruption of their intestinal microbiota or immune constitution to develop CDI (ie they don’t develop CDI from antibiotics alone without being exposed to CD in the environment)

Line 36- I don’t know that such granular information about the cost is completely related to you research question, especially since you weren’t attempting to compare LOS in the analysis.

Line 41-42- Fulminant disease severity does change the dose of vancomycin and this statement should be amended to acknowledge that.

Lines 43-57- I am still questioning why there is a focus on the history of vancomycin as a treatment when you aren't comparing various treatment options. I think all of this (and much of the next few paragraphs) could be consolidated by saying that now that vancomycin is a preferred treatment option for initial therapy in the 2017 treatment guidelines, there is more incentive than ever to establish the optimal dosing regimen.

Line 67-68- It’s unclear how this relates to the sentence before now that vancomycin is far more expensive now than in the 1980s? Wasn’t it always negligible compared to the cost of increased length of stay or readmission due to recurrence?

Line 72-74- it feels a little incongruent to say that the vancomycin recommendation was based on a high quality of evidence, but that the dose is not well elucidated. Although I agree with both statements individually, maybe it should be explained that the recommendation for vancomycin was based on high quality evidence, but that the dosage of vancomycin chosen in these comparative trials and thus proposed by the guidelines was not necessarily based on high quality evidence

Line 83-84- your statement about the conclusion of the study to give a higher loading dose isn’t supported by the information in the preceding sentences

Line 92- didn’t one of the studies relate concentrations to clinical outcomes? Not sure that we can say there was a complete lack of clinical correlation

Line 95- Would remove the statement about risk factors for recurrent CDI and therapies to reduce recurrence rate, since this doesn’t directly relate to data that you analyzed in the meta-analysis. In other words, if you had presented data about the number of patients on probiotics or rifaximin in these included studies I think it would make sense that you were including this sentence, but otherwise it doesn't seem to relate.

Methods- Although clinical cure was removed from meta-analysis, I feel like clinical cure should still be included in your methods since you set out to analyze that information and you pulled those studies. Then in the results you can say the meta-analysis wasn’t performed because of the heterogeneous definitions.

Methods- I would clarify that you looked only at initial non-fulminant CDI? And when you say comparing different dosing of vancomycin, which endpoints for comparison were you looking at? I would be more specific that you pulled studies that looked at either some marker of clinical cure and/or recurrence.

Statistical analysis- If the OR of recurrence is your secondary outcome, what is the primary outcome?

Line 146- Would remove the statement about not contacting authors, since this is implied by your methods

Results- It seems like the description of the studies in this section is just describing the dose that was given. I would allow the table to describe the specific dosing regimens and focus more on the intent of the studies. For example, it seems important to me to explain that although the Cunha study focused on adults that were not responsive to conventional oral vanco (which readers might confuse readers if they read the title of the paper), there was a small subgroup of patients who received high dose vanco empirically that were included in this analysis.

Figure 1- What are the other sources mentioned in that box? You only describe doing a database search in the methods. Maybe just remove this or explain what the other sources were?

Table 1- why is the Cunha study still included in the table if it’s not in the meta-analysis? As noted above, I do agree that it may be valuable to review other related studies that weren’t included in the meta-analysis specifically, either in the results (a summary table?) or maybe discussion. I also think it’s unclear how three studies were included in the recurrence meta-analysis when only one specifically lists that as an outcome examined. You could consider adding a separate column for “recurrence definition” that describes each study’s individual definition of recurrence.

Line 179- It doesn’t seem like the information you found in meta-analysis or PK studies of fecal concentrations support your concern that high dose vanco may be more effective than low dose vanco? This seems to be most supported by the clinical cure data that wasn't really formally presented.  I would also suggest that the first paragraph of the discussion be more focused on interpreting the results of your study, not introducing new ideas.

Lines 193-209- I’m having a hard time understanding how this fits into your findings? Maybe instead of presenting specifics of the individual results of these studies, you can consolidate this information and relate it to your findings “ie detectable serum concentrations of vancomycin have been noted after oral administration in XYZ patient populations”

Lines 220-226- As above, I’m finding this information to be very granular and it could probably be consolidated because it seems like your point is that high-dose vancomycin is more costly and also has the potential for detectable serum concentrations in certain patient populations, which makes the question of optimal dosing important.

Line 239- duration of symptoms?

Conclusion- The last lines seem to be introducing new ideas instead of simply concluding what you found

Table 2- This is great information, but it’s not clear what this number means.  I assume it’s the days to clinical cure? I would label what this value is
